# Physical Properties of Mineral Fibers Depending on the Mineralogical Composition

**DOI:** 10.3390/ma14206108

**Published:** 2021-10-15

**Authors:** David Bombac, Martin Lamut, Primož Mrvar, Brane Širok, Benjamin Bizjan

**Affiliations:** 1Faculty of Natural Sciences and Engineering, University of Ljubljana, 1000 Ljubljana, Slovenia; primoz.mrvar@ntf.uni-lj.si; 2The Slovenian Centre of Excellence for Space Sciences and Technologies, 1000 Ljubljana, Slovenia; martin.lamut@space.si; 3Faculty of Mechanical Engineering, University of Ljubljana, 1000 Ljubljana, Slovenia; brane.sirok@fs.uni-lj.si (B.Š.); benjamin.bizjan@fs.uni-lj.si (B.B.)

**Keywords:** mineral wool, fibers, chemical composition, mechanical properties, nanoindentation

## Abstract

A developed methodology for determining the physical properties of mineral fibers prepared from different input mixtures under the same spinning wheel conditions is described and discussed. Energy dispersive X-ray fluorescence spectroscopy was combined with simultaneous thermal analysis and thermogravimetry to study the mineralogical composition and typical melting and crystallization temperatures. The mechanical properties measured with nanoindentation were related to the mineralogical properties and the results obtained are in agreement with the literature. The developed methodology shows reliable performance and demonstrates the ability to study the mechanical properties of mineral fibers, their mineralogical composition, and thermal properties. The presented experimental methodology opens up the possibility of researching the mechanical properties of mineral fibers for the purpose of defining production recipes in the field of mineral thermal insulation materials.

## 1. Introduction

Mineral wool is an inorganic insulation material made of thin fibers with many industrial applications due to its excellent thermal and sound insulation properties and its good mechanical strength. It is most often produced by the fiberization of molten rock on the fast rotating wheels of a spinning machine [1,2]. A high quality fiberization process is essential to achieve the desired properties of the end products and is determined by long and thin fibers, a low proportion of unfiberized material and high mechanical strength.

The properties of mineral fibers depend on several design and operating parameters of spinning machines (e.g., wheel speeds and sizes, the wheel cooling system, properties of the blow-off flow, etc.) and on the melt properties (temperature, chemical composition, point of impact on the wheel, etc.) [1,3,4,5,6,7,8]. The fiber properties are usually determined in standardized tests carried out in the laboratories of mineral wool production plants. The most frequently investigated properties are the fiber diameter [9,10] and the tensile or compressive strength of the finished products—mineral wool panels [11,12]. Mechanical strength testing is usually performed only at the macroscopic level on samples consisting of many fibers. There is a highly complex relationship between the operating parameters of the manufacturing process and the macroscopic mechanical strength of mineral wool products. However, the favorable mechanical properties at the macroscopic level are determined by the properties of individual fibers (e.g., tensile strength, surface hardness, brittleness, etc.).

There have been only a few studies in which the mechanical properties of individual mineral fibers have been investigated. Gur’ev et al. [13] compared the tensile strength and modulus of elasticity of basalt fibers produced by different roving and blowing processes. Zhao et al. [14] investigated the influence of chemical composition on the tensile strength and corrosion resistance of rock, glass and slag wool fibers and showed a significant influence of melt composition. In addition to tensile strength, the modulus of elasticity [15,16,17] and hardness [16,17,18] of the fibers are also important properties that determine the mechanical properties of the finished mineral wool product. Due to the small fiber diameters (typically less than 10 µm), the measurement of these properties requires specialized measurement methods such as nanoindentation [19,20] or Sentmanat extensional rheometry [21].

The present work deals with the methodology and results of the study of fiber properties by nanoindentation, where the fibers were prepared under identical conditions but from melts with different chemical compositions. This study consists of two phases; in the first part the same operating conditions were applied to produce fibers from different input mixtures of diabase and blast furnace slag using a single wheel laboratory melt spinner. In the second part, a detailed study of the physical properties of the fibers produced is carried out with the aim of combining the thermodynamic properties of the melts of the input mixtures and the mechanical properties of the fibers with the chemical composition of the input mixture.

## 2. Materials and Methods

### 2.1. Laboratory Fibers Production

Four different premixed input mixtures of diabase and blast furnace slag were supplied for our study. The slag content in the premixed mixtures varied from 25 to 55 wt%, while the rest consisted of diabase. The exact chemical composition of diabase and blast furnace slag is unknown as input mixtures were supplied premixed for our study. The chemical compositions of the premixed input mixtures are listed in Table 1 and were determined by atomic absorption spectrometry at a temperature of 105 °C (PerkinElmer AAnalyst 600 (PerkinElmer, Waltham, MA, USA) and analyzed using PerkinElmer Analytical methods for atomic absorption spectrometry). The input mixtures were melted and spun into fibers on a laboratory single wheel melt spinner, with the experimental set-up shown in Figure 1 and Figure 2, always under the same operating conditions with the aim to eliminate the fiber production process as a possible cause of differences in the mechanical properties between fibers from different mixtures. In this way, the influence of chemical composition on the physical properties of fibers could be directly compared between different samples used in this study.

The spinning machine used for fiber production (Figure 2) consisted of a single rotating wheel with a radius of 95 mm. A three-phase electric motor was used to rotate the spinning wheel, where the speed of rotation was controlled by a variable frequency drive (VFD). The motor speed was kept at a constant frequency of 40 Hz. The spinning machine was in a circular housing with a diameter of 520 mm that was utilized to receive fibers, beads and other fiberization products that formed on the wheel.

A 20 kW induction furnace and a graphite crucible were used to prepare the melt because the input mixtures of diabase and blast furnace slag are electrically nonconductive. The crucible was heated by eddy currents generated by the surrounding water-cooled induction coil. Temperature measurements of the melts were carried out with a pyrometer. After stabilization at the temperature of 1450 °C for 300 s, the melt was poured from the graphite crucible into a V-shaped open melt channel with adjustable position and angle (Figure 2a) before flowing onto the spinning wheel. The melt flow hit the wheel at an angle of about 30° relative to the vertical centerline (Figure 1). To prevent the melt from solidifying in the channel, its surface was heated to about 450 °C with a gas burner. The melt was poured out by rotating the crucible using a swivel mechanism while maintaining a mass flow *Q*_m_ of about 0.15 kg·s^–1^ (Figure 2b). After impacting the single spinning wheel, part of the melt flow adhered to the wheel surface and formed a thin radial film, while the remaining melt was deflected by the wheel onto the housing wall (Figure 2b). For comparison, industrial spinning machines consist of multiple (2–4) wheels, and a part of the melt is diverted in a cascade flow from the first to the second wheel, etc., until the remaining melt becomes fiberized at the last wheel [1].

After completion of the fiberization process, part of the fibers accumulated in the spinning machine housing were taken for chemical, thermodynamic, and nanoindentation analyses.

### 2.2. Methodology for Physical Properties Analysis of Fibers

The properties of the fibers produced depend on a variety of material and processing parameters. By using changes in input mixtures to fine-tune the chemical composition of the melt, fibers with the desired physical properties and targeted costs can be produced. With the laboratory spinning wheel developed by the authors, we can produce fibers with the same processing parameters, varying only the input mixture used.

In order to connect the physical properties of fibers with the input mixture, we have developed a methodology for experimental analysis in which relevant fiber properties are related to the input mixture. Within the developed methodology, the chemical composition of the fibers, the relevant transition temperatures and finally the mechanical properties of the fibers are investigated. A detailed description of the experimental methods used can be found below.

Chemical composition of produced fibers was determined by energy dispersive X-ray fluorescence (XRF), since this method can be used to quantify or qualify almost any element that should be present in the fibers or glassy melt. The XRF analyzer used was Thermo Fisher Scientific NITON model XL3t + GOLDD-900S He (Thermo Fisher Scientific, Waltham, MA, USA), with a He gas purging device, a silver anode, and a 50 kV generator. Prior to measurement, fibers and bulk glass residues were pulverized in a ball mill and the sampling site was purged with He gas for better detection of lighter elements (Mg, Si, Al, P). The measuring time for each sample was 300 s. To assess the accuracy and precision of the measurements, precalibration measurements against 24 international standards (National Institute of Standards and Technology (NIST) and United States Geological Survey (USGS)) and a calibration with NIST-1d and NIST-88b were performed at the beginning and at the end. A statistical analysis (PCA—principal component analysis) was performed with Statistica (version 8.0) and Grapher (version 10.0). In order to ensure accurate and repeatable measurements and to better determine the relationship between elemental variability between samples, an analysis of duplicate samples was performed.

Melting and crystallization temperatures and the corresponding enthalpies of the input mixtures were evaluated experimentally with differential scanning calorimetry (DSC). The DSC was performed on a STA 449 Jupiter instrument from NETZSCH (Netzsch Holding, Selb, Deutschland), with all analyses being performed at ambient pressure and under a protective argon atmosphere. The experiments were performed at heating/cooling rates of 10 °C·min^–1^, according to the heat flux mode. In this mode, the investigated and comparison samples were heated with the same heat source, using an empty graphite crucible as reference sample. Prior to the DSC measurements of the fibers, a baseline measurement was performed with empty graphite crucibles. The baseline curve was used to correct the DSC signal of the samples.

The optimal diameter of the fibers is less than 10 µm, so it is difficult to determine the mechanical properties of the fibers. One of the possible methods to determine the mechanical properties of a particular fiber is nanoindentation. The mechanical properties of the fibers produced from different input mixtures were determined in our case by hardness testing on a series of produced fibers under static loading conditions using an Agilent G200 nanoindenter (Agilent Technologies, Santa Clara, CA, USA). Fibers from each input mixture were embedded in epoxy resin and multiple indentations were made using a three-sided diamond Berkovich indenter tip (E = 1141 GPa and ν = 0.07). The apparent radius of the indenter tip is about 100 nm, so the indentation depth during measurement must be more than 100 nm to avoid deviation from the calculated projected area of the indentation and correlated size effects. The continuous stiffness measurement (CSM) method was used, which allows dynamic indentation to observe the mechanical properties as a function of penetration into the surface of the sample. Several measurements were made on a set of fibers embedded in epoxy resin. The obtained force-displacement curves were analyzed using the method of Oliver and Pharr [22].

The presented chain of fiberization and the subsequent investigations of physical properties made it possible to determine important properties of individual produced fibers and to investigate the differences between them. In addition, the investigations carried out on the different input mixtures allowed us to point out the differences and optimal properties of the produced fibers in relation to the input mixture of diabase and blast furnace slag.

## 3. Results and Discussion

The chemical composition of the produced fibers as determined by XRF is given in Table 2, together with the results of the SiO_2_ reference sample. The results show that silica (SiO_2_) is the main component in all samples. Depending on the input mixture, the silica content was between 47 and 49 wt% and Alumina (Al_2_O_3_) with a content between 11.5 and 13 wt%. The combined SiO_2_ + Al_2_O_3_ content in the fibers for this study ranged from about 57 to 62 wt%. In the published literature (e.g., [21]), the combined content of SiO_2_ + Al_2_O_3_ is referred to as the ceramic content.

Compared to E-glass or basalt fibers, the combined content of silica and alumina in the fibers prepared for this study was lower for approximately 10 to 20 wt% [21,23]. In slag fibers, large amounts of calcium oxide (CaO), magnesium oxide (MgO) and iron oxide (Fe_2_O_3_) are typical [6,24], and were measured in the range between 32 and 37 wt%. CaO, MgO and other components are known as network modifiers that break the SiO_2_ and Al_2_O_3_ network structure [25]. Other detected constituents were manganese oxide (MnO), titanium dioxide (TiO_2_), potassium oxide (K_2_O) and a minor amount of chromium oxide (Cr_2_O_3_). The chemical composition of the fibers in our study varied mainly in terms of the content of CaO and to a lesser extent in terms of the content of SiO_2_ and Fe_2_O_3_. The XRF also allowed the determination of trace elements contained in the samples. The results of the analysis of the trace elements are shown in Table 3, where S, Cu, Ni, Sr, Zr and Ba were present in relatively large quantities. Trace elements detected in relatively small amounts were Pb, Rb, Sr, Zn and Nb. The remaining nondetected elements were trapped gases and Na_2_O, which are beyond the resolution of the XRF equipment used.

The transition and crystallization temperatures of the produced fibers were determined with DSC in combination with thermogravimetric measurements (TG, dash dotted lines in Figure 3). The heating DSC thermogram is shown in Figure 3 as the change in heat flow rate with temperature. The first endothermic peak at about 100 °C was due to the removal of water present in the samples. The first exothermic peak was broad and started at a temperature of about 200 °C and was due to the oxidation of glass by the formation of metal oxides. The mechanism of oxidation is explained using the thermogravimetry results in the next paragraph. The small endothermic peak at a temperature of about 580 °C was attributed to the release of CO_2_ from the slag contained in the input mixture and was in agreement with the literature [26]. The glass transition was attributed to the endothermic peak at about 720 °C. The exothermic peak between temperatures of about 860 to 920 °C and the exothermic peak around 1000 °C were due to crystallization of the constituent phases. The glass transition peak and the onset of crystallization were in agreement with published results for diabase [27]. The peaks corresponding to the crystallization of diabase glass have been identified in the literature. The crystallization of diopside (CaO·MgO·2SiO_2_) occurs at about 865 °C and anorthite (CaO·AI_2_O_3_·2SiO_2_) at temperatures around 1060 °C [27]. The discrepancies in peak temperatures compared to the published literature were due to the higher heating rate used in the DSC measurements in the present study and the different CaO/MgO ratio in the chemical compositions as reported in the literature [28]. Therefore, the first crystallization peak was most likely due to the crystallization of diopside, while the crystallization of anorthite was responsible for the second crystallization peak. The temperature window of melting is very wide and depends strongly on the composition of the input mixture. The input mixtures for the fibers investigated contained different amounts of slag, the melting of which can be seen in the thermogram at about 1210 °C for all four fiber specimens investigated. From the increasing height of the peak, we can confirm that the amount of slag increased from the mixture labeled S1 up to S4, with S4 containing the highest amount of the slag.

Along the thermogram in Figure 3, the dash dotted line also shows the thermogravimetric measurements during heating. Thermogravimetric (TG) data is given as a normalized mass change Δ*m*/*m*_0_, where Δ*m* is the mass change and *m*_0_ is the initial mass of the sample. The increase in normalized mass was attributed to the oxidation of the fibers during the heating stage. Oxygen is incorporated into fibers and forms metallic oxides, where oxidation of Fe^2+^ to Fe^3+^ occurs [29,30,31]. Moreover, this behavior was indicated by a broad peak up to the temperature of the onset of crystallization. After the first exothermic crystallization peak, the mass of the sample began to decrease. The mass loss increased after the components began to melt and burn away at approximately 1200 °C. The cooling thermogram and thermogravimetric measurements are shown in Figure 4. Solidification started at about 1500 °C and ended between 1286 and 1310 °C, depending on the slag content of the sample. Details of the characteristic melting and solidification temperatures are given in Table 4. From the thermogravimetric measurements, the amount of mass loss depended on the slag content of the sample. Samples S1 and S2 were very close to each other and after heating and cooling cycles in DSC apparatuses, had about 5% lower mass. For samples S3 and S4, the amount of mass loss was 8% and 9%, respectively.

The measured hardness of a series of fibers prepared from different input mixtures at an indentation depth of 200 nm is shown in Table 5. Several measurements were made on the fibers from each input mixture and the results are presented as an average of all the measurements (Table 5). The method of Oliver and Pharr [19] used to analyze force-displacement data among other requirements assumes that the specimen is a flat semi-infinite half-space in which the stresses vanish at large intervals. In the present case, this was not the case, due to the small fiber radius (5–15 µm). However, Lonnroth et al. [16] evaluated the contribution of specimen geometry on the measured stiffness of the indenter and fiber contact. They showed that the magnitude of the error is a function of the indentation depth and the fiber radius, with an error of about 4% for a 150 nm indentation depth on a 2.5 µm diameter fiber. On this basis, the hardness and modulus of elasticity in our cases were determined at a penetration depth of 200 nm for all measurements in order to eliminate size effects and to avoid environmental influences such as the influence of the fiber-embedding epoxy substrate on the measured nanoindentation values. The force required to reach this penetration depth was between 3 mN and 4 mN for all fiber samples from specific input mixtures.

To the authors’ knowledge, there are no comparable single fiber microhardness studies using nanoindentation or other methods. Nevertheless, Deng et al. [14] investigated the Vickers hardness of solid aluminosilicate glasses (CaO-Al_2_O_3_-SiO_2_), which were reasonably close to the chemical composition of our fibers. The fiber surface hardness of our measurements, shown in Table 5, agree well with the results reported by Deng et al. [14] (approximately 6% difference), suggesting that nanoindentation provides plausible microhardness measurements. The discrepancy can be partially attributed to the difference in the hardness measurement method (nanoindentation vs. Vickers indentation) and chemical composition. Although the two hardness methods cannot be directly compared, they can be used as a measure in the order of magnitude of the fiber hardness as a rule of thumb. The hardness results obtained and the differences between individual fibers depend strongly on the amount of SiO_2_, Al_2_O_3_ and CaO. The chemical composition of the fibers given in Table 2 shows that the sum of all SiO_2_, Al_2_O_3_ and CaO was practically the same in all samples. However, their ratios were different and affected the fiber hardness. In sample S1, the proportion of SiO_2_ and Al_2_O_3_ was the highest, while sample S2 had the highest hardness, where a higher proportion of CaO was measured. Comparing the results with those of Deng et al. [17], the sample with the lowest SiO_2_ content, which also had the highest CaO content, should have the highest hardness. The Al_2_O_3_ content was practically the same in samples S1, S2 and S4, while it was slightly lower in sample S3, which affected the measured hardness values.

Sample preparation for nanoindentation requires careful execution in terms of sample size and shape. Sample preparation for nanoindentation of fibers was not straightforward to avoid scatter in the nanoindentation measurement results. The nanoindentation samples were prepared for the initial measurements by placing the fibers on a metal holder and then embedding them in epoxy resin. Prior to measurement, standard metallurgical sample preparation was performed by grinding and polishing. This preparation procedure resulted in poor convergence because the condition of the fibers was not known during the sample preparation. The preparation procedure may have resulted in broken or partially cracked fibers, or there may have been air pockets under the fibers, all leading to scattered results. Improvement in the scatter of measurement data was achieved by changing the sample preparation method, in which fibers were randomly placed in an epoxy resin. Upon grinding and polishing, it was found that some of the fibers on the surface were fully embedded in the resin and the results converged in a narrower band.

Figure 5 shows a typical plot of hardness as a function of indenter penetration depth in a series of measurements on fibers from input mixture S1. The analysis was performed at a depth of 200 nm, as explained earlier, to reduce various effects on the results. The influence of the environment (fiber embedding epoxy resin substrate) during the measurement can be seen as a constant decrease in the measured values with increasing penetration depth. Considering the results of Lonnroth et al. [16] and their use of geometric correction, as well as comparison with the fiber sizes and nanoindentation penetration depth used in this study, it was decided not to apply geometric correction to the stiffness calculation. Comparison of the fiber sizes and nanoindentation penetration depth used in this study, it was decided not to apply the geometric correction to the stiffness calculation. The maximum error reported by Lonnroth et al. [16] was about 4%, which was close to the standard deviation of the obtained results in our case. Moreover, the purpose of determining the mechanical properties of fibers with nanoindentation was to discover trends in the changes based on different input mixtures.

Figure 6 shows a typical plot of the modulus of elasticity as a function of penetration depth. The average measured values of the modulus of elasticity at a penetration depth of 200 nm are given in Table 6. Even at smaller penetration depths, e.g., 200 nm, the results scattered considerably, which can be explained by the much larger plastic zone compared to the elastic zone included in the hardness measurements. A larger plastic zone means that more environment (fiber embedding epoxy resin substrate) is considered in the calculation of the modulus. Therefore, for thin specimens, the substrate is very quickly considered when measuring the elastic response. The same mechanism can be used to explain the significant decrease in modulus values. The effects of the substrate were reduced by choosing smaller penetration depths. A comparison with the results in Oseli et al. [17] showed agreement to the expected extent (about 3–12% difference). It should be noted that results of Oseli et al. [17] were adapted to different fibers (E-glass fibers, basalt fibers, melted mineral wool fibers) and a fitted line for the combined content of SiO_2_ + Al_2_O_3_ as a function of modulus of elasticity was constructed. The values fitted to the chemical composition of the present study were used to compare the values with the results of modulus of elasticity in the present study.

## 4. Conclusions

In this study, the thermodynamical and mechanical properties of fibers made with the same processing parameters with a single wheel laboratory spinner were investigated with the aim to connect the chemical composition of the input mixtures to the mechanical properties determined with nanoindentation. Moreover, the chemical composition of the produced fibers was connected also to the relevant melting and crystallization temperatures. The following conclusions can be drawn from the results obtained:The variation of diabase and iron slag content in the input mixture for melt fiberization has a significant effect on melting and crystallization temperatures.Nanoindentation is suitable for measuring both surface hardness and elastic modulus of individual fibers.The surface hardness of fibers measured by nanoindentation is in good agreement with bulk measurements on glass of similar composition reported in the literature.Nanoindentation revealed that the highest hardness was obtained in the sample S2.The modulus of elasticity measured by nanoindentation agrees well with the values reported in the literature, which were adjusted to the input mixtures from the present study.

## Figures and Tables

**Figure 1 materials-14-06108-f001:**
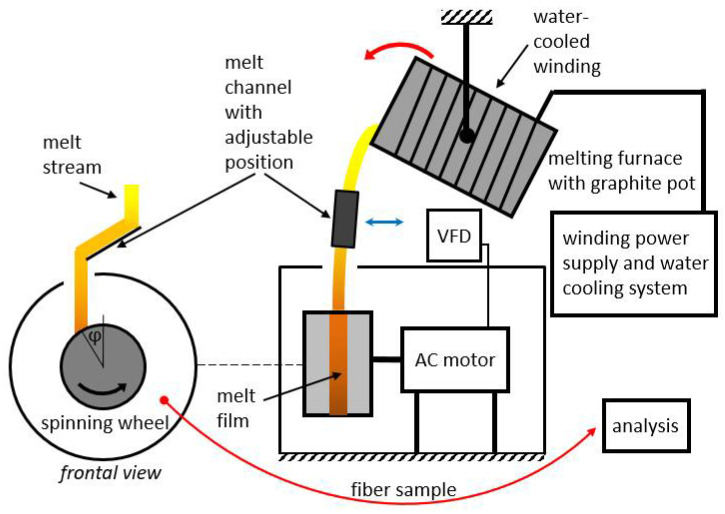
Experimental fiber production set-up.

**Figure 2 materials-14-06108-f002:**
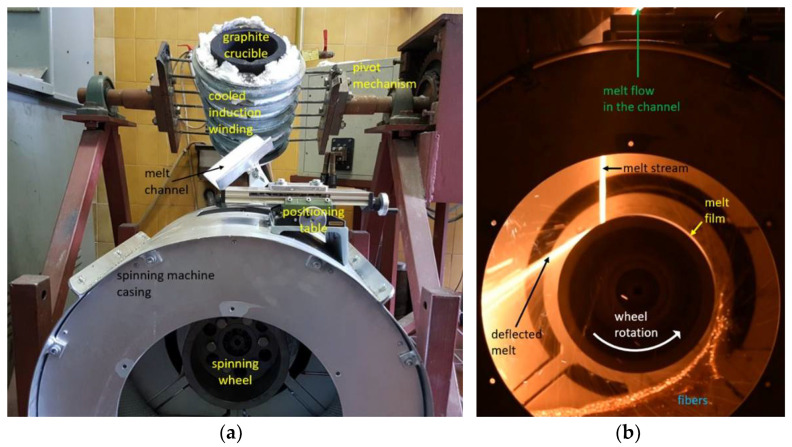
Laboratory single wheel spinning machine for melt fiberization; (**a**) experimental setup with induction furnace and (**b**) melt flow onto a wheel and fiberization.

**Figure 3 materials-14-06108-f003:**
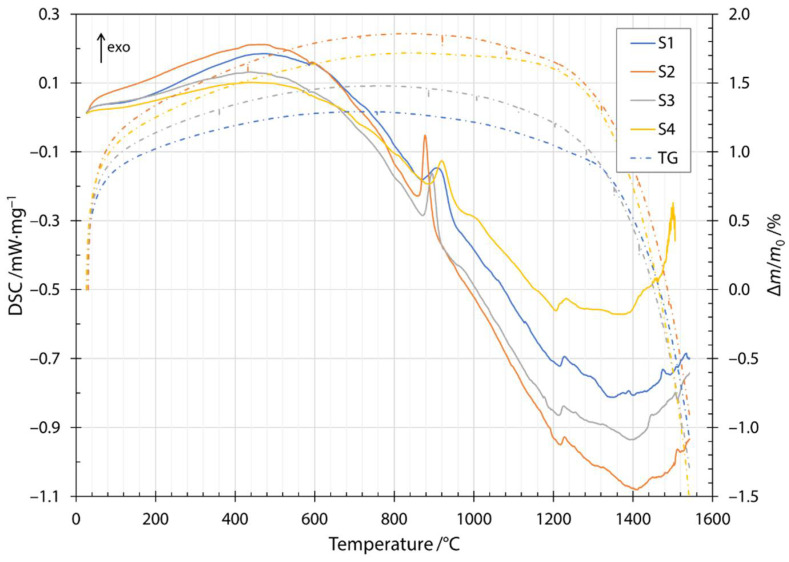
Heating DSC thermogram and thermogravimetry for samples S1, S2, S3 and S4.

**Figure 4 materials-14-06108-f004:**
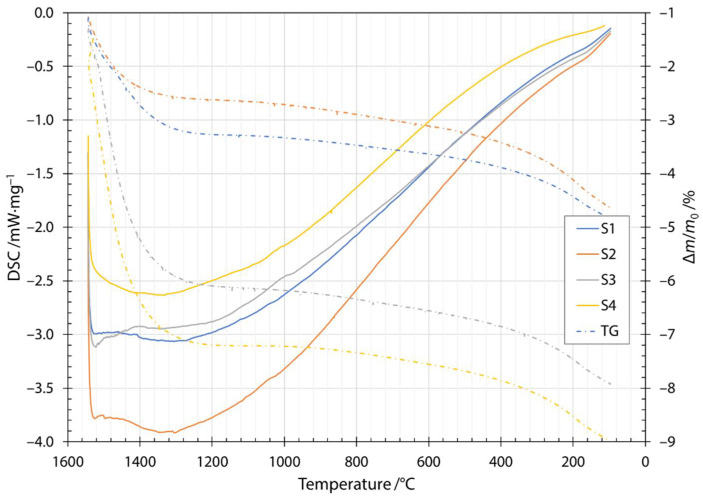
Cooling DSC thermogram and thermogravimetry for samples S1, S2, S3 and S4.

**Figure 5 materials-14-06108-f005:**
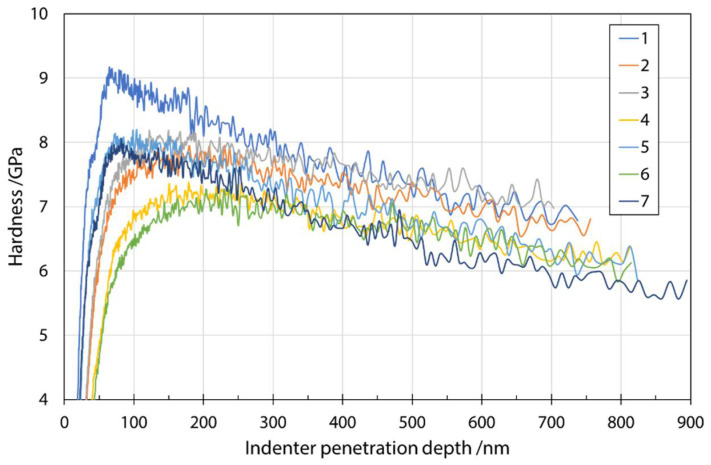
Hardness as a function of indenter penetration depth for input mixture S1.

**Figure 6 materials-14-06108-f006:**
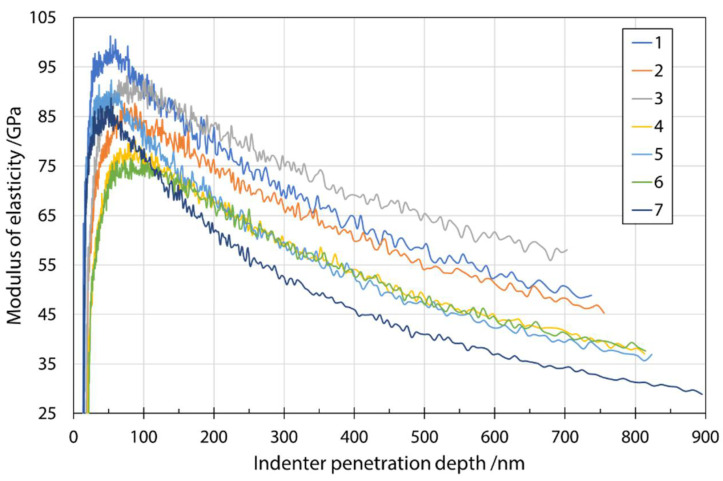
Modulus of elasticity as a function of indenter penetration depth for input mixture S1 determined during nanoindenter unloading.

**Table 1 materials-14-06108-t001:** Chemical composition of input mixtures in wt%.

	S1	S2	S3	S4
SiO_2_	44.77	45.47	46.02	47.72
Al_2_O_3_	13.6	13.2	12.2	12.3
Fe_2_O_3_	4.75	4.1	3.46	3.06
MgO	7.95	8.13	7.9	7.47
CaO	18.3	18.6	20.95	22.28
K_2_O	0.44	0.45	0.55	0.57
TiO_2_	1.09	0.94	0.84	0.72
MnO	0.99	1.08	1.56	1.64
FeO	2.15	2.83	2.16	1.82
Na_2_O	2.97	2.57	2.37	0.93
Volatiles/LOI *	2.99	2.63	1.99	1.49

* Loss on ignition.

**Table 2 materials-14-06108-t002:** Chemical composition of studied fibers and SiO_2_ reference sample in wt%.

	S1	S2	S3	S4	SiO_2_
SiO_2_	49.0	47.48	46.01	46.92	99.90
Al_2_O_3_	13.0	12.29	11.59	12.21	0.46
Fe_2_O_3_	6.6	5.6	3.86	4.73	0.02
MgO	10.36	10.2	9.93	9.91	
CaO	15.95	18.52	22.7	20.71	0.01
K_2_O	0.42	0.49	0.64	0.55	0.27
TiO_2_	0.87	0.82	0.66	0.72	0.04
MnO	0.95	1.26	1.72	1.5	
Cr_2_O_3_	0.04	0.03	0.03	0.03	

**Table 3 materials-14-06108-t003:** Trace elements in analyzed samples as determined by XRF in ppm.

	S1	S2	S3	S4
Cu	54.96	152.65	44.32	79.63
Nb	5.92	8.06	11.04	9.80
Ni	119.24	113.14	88.18	75.14
Pb		8.28		
Rb	12.93	15.78	22.41	19.07
Sr	274.21	314.64	383.66	350.68
Zn	77.42	171.62	23.85	52.42
Zr	72.55	80.38	82.41	81.41
Ba	748.02	968.31	1262.69	1112.5
S	1596.2	2411.7	3570.1	2938.7

**Table 4 materials-14-06108-t004:** Characteristic melting and crystallization temperatures of different input mixtures as determined by DSC measurements.

Sample	Heating (Melting)		Cooling (Solidification)	
	*T*_min_/°C	*T*_max_/°C	*T*_min_/°C	*T*_max_/°C
S1	1300	1468	1408	1310
S2	1336	1504	1501	1309
S3	1324	1507	1441	1289
S4	1292	1499	1430	1286

**Table 5 materials-14-06108-t005:** Average of measured hardness for different input mixtures and comparison to literature.

Sample	Hardness/GPa	
	Present Study	Deng et al. [14]
S1	7.68 ± 0.35	6.95
S2	7.76 ± 0.38	7.05
S3	7.10 ± 0.35	7.12
S4	7.55 ± 0.38	7.09

**Table 6 materials-14-06108-t006:** Average of measured modulus of elasticity for different input mixtures and comparison to literature.

Sample	Modulus of Elasticity/GPa	
	Present Study	Oseli et al. [18]
S1	73.2 ± 8.5	82.1
S2	70.9 ± 3.8	71.8
S3	60.5 ± 6.1	58.5
S4	64.9 ± 6.6	66.6

## Data Availability

The data presented in this study are available on request from the corresponding author. The data are not publicly available due to commercial interest of input material supplier.

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
