# Peer review of "Physical Properties of Mineral Fibers Depending on the Mineralogical Composition"

_materials, 2021, doi:10.3390/ma14206108_

Round 1
Reviewer 1 Report
Revision of the manuscript “Mineral fibers physical properties dependent on the mineralogical composition” by Bombac et al. submitted for publication to Materials.
The authors describe a method to produce and characterise at a laboratory scale rock wool. The only innovative content of the ms is represented by the measurement of harness on single fibers through nanoindentation experiments. Concepts such as rock, mineral and chemical composition are used improperly and interchangeably, whereas they are not. The authors are invited to learn those concepts and revise the ms consistently. The paragraph n. 2 is named “Materials and Methods”, but there is not a sufficient description of the starting material. I discovered just at the end that it could be diabase. The provided XRF chemical composition of the starting materials is incomplete. It needs to be complemented with the volatile content determined through loss on ignition experiments and also the slag amount of the starting slag-diabase mixtures need to be provided. The discussion of the DSC and TG results versus phase transitions need to be improved. The English text is sometimes not comprehensible and require correction/rewording. These and other minor changes and suggestions are detailed below. The paper may be worthy of publication after this major revision.
Line 65 and table 1. Mineralogical composition and chemical composition, even if related, are different concepts: there is the same relationship/difference as for menu and ingredients! That reported in table 1 are chemical components. Mineral composition is made of phases (quartz, calcite, rutile...). Please, get informed and revise consistently.
Table 1. Usually, chemical compositions are complemented with volatiles, H2O, CO2… determined through loss on ignition experiments, and overall should sum up to about 100%. Please, provide with the missing information.
Line 68: same confusion between mineralogy and chemistry as above.
Line 81: “single wheel spinning wheel“, repetition?
Line 106 and throughout the text: a mineral by definition is produced by natural processes, is crystalline and possess a defined composition. Those described in this paper are not mineral, but synthetic glasses.
Line 110: “…varies is only…” revise English.
Line 112-113: replace “combined” with “related”.
Line 115: “each” specify what.
Line 118. XRF can determine only the chemical composition, not the mineralogical composition. See above.
Line 136 and 137: replace “comparison” with “reference”.
Line 160: somewhere you use “mineral input mixture”, here and somewhere else “input rock mixture”. Again, mineral and rock, although related, are different concepts. There is the same difference/relationship as for goods and supermarket! Please, get informed and revise consistently.
Line 163: see above comments.
Line 164: “component”.
Line 166, 167… and similar throughout the text: CaO is not a mineral in this case, is just a chemical component of a material. Revise the caption of table 2.
Line 164-179, 237 and throughout the text: oxide stoichiometric indexes should be subscripts.
Line 182: specify which “mineral constituent” crystallises.
Line 183: specify which “mineral constituent” begins to melt.
Line 185: diabase is a rock with a defined chemical and mineralogical composition. It should not be difficult to relate DSC peaks with specific phase transitions.
Line 188-190: the amount of slag should be known from the input material, since has a different nature and provenance respect the diabase rock, so why to estimate it from the DSC data? It would be more logic to discuss input slag amounts vs DSC peaks.
Line 227: “environmental influences”? Please, specify/revise.
Figure 5: can be omitted, does not add any additional information that is not given in table 4.
Line 243: “…hardness measurement methods…”.
Line 244: “…as a measure of the order of magnitude of the fiber hardness”.
Line 245-246: not comprehensible…
Line 246-247: “…the first set of samples for nanoindentation measurements…”
Line 253: “A reduction of the scattering of data…”.
Line 254: “grinding” after embedding in epoxy? May be “cutting”?
Line 262: which “environment” you are referring to? Please, clarify.
Line 263-266: explain why. The reason is resumed in the sentence that follows? In that case reword text.
Line 267-269: it appears like a justification for a missed discussion. Please, reword/clarify.
Figure 7. Wrong y-axis label.
Line 281: for “environment” you mean confining medium? If yes, use the latter here and above.
Line 286-288: not comprehensible…
Line 292: “chemo-geological”, that is new! What is “geological” in this paper?
Author Response
We are grateful for the comments (italic), which have been addressed as follows:
Concepts such as rock, mineral and chemical composition are used improperly and interchangeably, whereas they are not. The authors are invited to learn those concepts and revise the ms consistently.
Apologies, only chemical composition is used in revised manuscript.
The paragraph n. 2 is named “Materials and Methods”, but there is not a sufficient description of the starting material. I discovered just at the end that it could be diabase. The provided XRF chemical composition of the starting materials is incomplete. It needs to be complemented with the volatile content determined through loss on ignition experiments and also the slag amount of the starting slag-diabase mixtures need to be provided.
In the revised manuscript description of the starting material was added along to chemical composition determined by atomic absorption spectrometry. Following text was added to revised manuscript: "Four different premixed input mixtures of diabase and blast furnace slag were supplied for our study. The slag content in the premixed mixtures varied from 25 to 55 wt %, while the rest consisted of diabase. The exact chemical composition of diabase and blast furnace slag is unknown as input mixtures were supplied premixed for our study. The chemical compositions of the premixed input mixtures are listed in Table 1 and were determined by atomic absorption spectrometry at a temperature of 105 °C (Perkin Elmer AAnalyst 600 and analyzed using Perkin Elmer Analytical methods for atomic absorption spectrometry)."
The discussion of the DSC and TG results versus phase transitions need to be improved.
DSC and TG results were expanded and discussed considering available literature. In the revised manuscript this section now reads as: "Transition and crystallization temperatures of the produced fibers were determined with DSC in combination with thermogravimetric measurements (TG, dash dotted lines in Figure 3). The heating DSC thermogram is shown in Figure 3 as the change in heat flow rate with temperature. The first endothermic peak at about 100 °C is due to the removal of water present in the samples. The first exothermic peak is broad and starts at a temperature of about 200 °C and is due to the oxidation of glass by the formation of metal oxides. The mechanism of oxidation is explained using the thermogravimetry results in the next paragraph. The small endothermic peak at a temperature of about 580 °C was attributed to the release of CO2 from the slag contained in the input mixture and is in agreement with the literature [26]. The glass transition was attributed to the endothermic peak at about 720 °C. The exothermic peak between temperatures of about 860 to 970 °C and the peak around 1010 °C are due to crystallization of the constituent phases. The glass transition peak and the onset of crystallization are in agreement with published results for diabase [27]. The peaks corresponding to crystallization in diabase have been identified in the literature. Crystallization of diopside (CaO·MgO·2SiO2) occurs at temperatures corresponding to the first crystallization peak, while crystallization of anorthite (CaO·AI2O3·2SiO2) occurs at temperatures around 1060 °C [27]. Therefore, the first crystallization peak is due to the crystallization of diopside, while the crystallization of anorthite is responsible for the second crystallization peak. The temperature window of melting is very wide and depends strongly on the composition of the input mixture. The input mixtures for the fibers investigated contained different amounts of slag, the melting of which can be seen in the thermogram at about 1210 °C for all four fiber specimens investigated. From the increasing height of the peak, we can confirm that the amount of slag increased from the mixture labeled S1 up to S4, with S4 containing highest amount of the slag.
Along the thermogram in Figure 3, the dash dotted line also shows the thermogravimetric measurements during heating. Thermogravimetric (TG) data is given as a normalized mass change Δm/m0, where Δm is the mass change and m0 is the initial mass of the sample. The increase in normalized mass was attributed to the oxidation of the fibers during heating stage. Oxygen is incorporated into fibers and forms metallic oxides, where oxidation of Fe2+ to Fe3+ occurs [28–30]. Moreover, this behavior is indicated by a broad peak up to the temperature of the onset of crystallization. After the first exothermic crystallization peak, the mass of the sample began to decrease. The mass loss increased after the components began to melt and burn away at approximately 1200 °C. Cooling thermogram and thermogravimetric measurements are shown in Figure 4. Crystallization starts at about 1500 °C and ends between 1286 and 1310 °C, depending on the slag content of the sample. Details of the characteristic melting and crystallization temperatures are given in Table 3. From the thermogravimetric measurements amount of the mass loss depends on the slag content of the sample. Samples S1 and S2 are very close to each other and after heating and cooling cycles in DSC apparatuses have about 5 % lower mass. For samples S3 and S4, the amount of mass loss is 8 % and 9 %, respectively."
The English text is sometimes not comprehensible and require correction/rewording.
English language was revised.
Line 65 and table 1. Mineralogical composition and chemical composition, even if related, are different concepts: there is the same relationship/difference as for menu and ingredients! That reported in table 1 are chemical components. Mineral composition is made of phases (quartz, calcite, rutile...). Please, get informed and revise consistently.
Apologies, corrected.
Table 1. Usually, chemical compositions are complemented with volatiles, H2O, CO2… determined through loss on ignition experiments, and overall should sum up to about 100%. Please, provide with the missing information.
Table 1 was replaced with chemical composition of received premixed mixtures determined with atomic absorption spectrometry.
Line 68: same confusion between mineralogy and chemistry as above.
Apologies, corrected.
Line 81: “single wheel spinning wheel“, repetition?
Apologies, corrected.
Line 106 and throughout the text: a mineral by definition is produced by natural processes, is crystalline and possess a defined composition. Those described in this paper are not mineral, but synthetic glasses.
Line 110: “…varies is only…” revise English.
Apologies, corrected.
Line 112-113: replace “combined” with “related”.
Apologies, corrected.
Line 115: “each” specify what.
Apologies, corrected.
Line 118. XRF can determine only the chemical composition, not the mineralogical composition. See above.
Apologies, corrected.
Line 136 and 137: replace “comparison” with “reference”.
Apologies, corrected.
Line 160: somewhere you use “mineral input mixture”, here and somewhere else “input rock mixture”. Again, mineral and rock, although related, are different concepts. There is the same difference/relationship as for goods and supermarket! Please, get informed and revise consistently.
Apologies, corrected.
Line 163: see above comments.
Apologies, corrected.
Line 164: “component”.
Apologies, corrected.
Line 166, 167… and similar throughout the text: CaO is not a mineral in this case, is just a chemical component of a material. Revise the caption of table 2.
Apologies, corrected.
Line 164-179, 237 and throughout the text: oxide stoichiometric indexes should be subscripts.
Apologies, corrected.
Line 182: specify which “mineral constituent” crystallises.
Line 183: specify which “mineral constituent” begins to melt.
Line 185: diabase is a rock with a defined chemical and mineralogical composition. It should not be difficult to relate DSC peaks with specific phase transitions.
Line 188-190: the amount of slag should be known from the input material, since has a different nature and provenance respect the diabase rock, so why to estimate it from the DSC data? It would be more logic to discuss input slag amounts vs DSC peaks.
Line 227: “environmental influences”? Please, specify/revise.
Apologies, corrected.
Following text was used in revised manuscript: "On this basis, the hardness and modulus of elasticity in our cases were determined at a penetration depth of 200 nm for all measurements in order to eliminate size effects and to avoid environmental influences such as the influence of the fiber-embedding epoxy substrate on the measured nanoindentation values."
Figure 5: can be omitted, does not add any additional information that is not given in table 4.
Figure 5 was removed from revised manuscript.
Line 243: “…hardness measurement methods…”.
Apologies, corrected.
Line 244: “…as a measure of the order of magnitude of the fiber hardness”.
Apologies, corrected.
Line 245-246: not comprehensible…
Apologies, corrected.
Line 246-247: “…the first set of samples for nanoindentation measurements…”
Apologies, corrected.
Line 253: “A reduction of the scattering of data…”.
Apologies, corrected.
Line 254: “grinding” after embedding in epoxy? May be “cutting”?
Samples were only ground after embedding in epoxy resin.
Line 262: which “environment” you are referring to? Please, clarify.
As environment it was meant epoxy substrate. In revised manuscript following was added after environment throughout the text "(fiber embedding epoxy resin substrate)".
Line 263-266: explain why. The reason is resumed in the sentence that follows? In that case reword text.
Apologies, corrected.
Line 267-269: it appears like a justification for a missed discussion. Please, reword/clarify.
Apologies, corrected.
Figure 7. Wrong y-axis label.
Apologies, corrected.
Line 281: for “environment” you mean confining medium? If yes, use the latter here and above.
That was meant as environment. Same correction as for comment "Line 262" was used.
Line 286-288: not comprehensible…
Apologies, corrected.
Line 292: “chemo-geological”, that is new! What is “geological” in this paper?
Apologies, corrected.
Reviewer 2 Report
1.Some recent published papers related to mineral wool should be reviewed in the section of "1. Introduction".
[1] Adjusting the melting and crystallization behaviors of ferronickel slag via partially replacing of SiO2 by B2O3 for mineral wool production. Waste Management 2020, 111: 34–40 [2] Effect of B2O3 on the properties of ferronickel melt and mineral wool [J]. Ceramics International, 2020, 46 (9): 13460-13465.
[3] Viscosity and Structure of MgO-SiO2-based Slag Melt with Varying B2O3 Content. Ceramics International, 2020, 46(3): 3631-3636.
2.Have you evaluated the effect of graphite crucible on chemical composition of the raw materials since the raw materials contains Fe2O3 and MnO, Cr2O3, which can be reduced at the reducing atmosphere.
3.Line 98, page 3, "typically with 2-4 wheels, ", how many wheels in your lab equipment?
4.Table 2, what is the unit of those value? It is very confusing if it is compared with Table 1. In table, the component are oxides, but here they are elements.
5.Lines 199-200, page 6, "The increase in normalized mass was attributed to the oxidation of the fibers during heating stage. " So, what is oxidized during the heating stage?
6.Table 5, the change of modulus of elasticity and hardness with the variation of composition of those mixture, and why they changed like that should be stated in the text.
Author Response
We are grateful for the comments (italic), which have been addressed as follows:
1.Some recent published papers related to mineral wool should be reviewed in the section of "1. Introduction".
[1] Adjusting the melting and crystallization behaviors of ferronickel slag via partially replacing of SiO2 by B2O3 for mineral wool production. Waste Management 2020, 111: 34–40 [2] Effect of B2O3 on the properties of ferronickel melt and mineral wool [J]. Ceramics International, 2020, 46 (9): 13460-13465.
[3] Viscosity and Structure of MgO-SiO2-based Slag Melt with Varying B2O3 Content. Ceramics International, 2020, 46(3): 3631-3636.
Suggested references were included into revised manuscript.
2.Have you evaluated the effect of graphite crucible on chemical composition of the raw materials since the raw materials contains Fe2O3 and MnO, Cr2O3, which can be reduced at the reducing atmosphere.
Reduction of graphite crucible was evaluated only with simple magnetic test. Since glass at the end did not exhibit magnetic properties, we concluded not to further evaluate reduction of crucibles.
3.Line 98, page 3, "typically with 2-4 wheels, ", how many wheels in your lab equipment?
Only one spinning wheel is used in laboratory spinner. Explanation was added in line 102 in the revised manuscript.
4.Table 2, what is the unit of those value? It is very confusing if it is compared with Table 1. In table, the component are oxides, but here they are elements.
Apologies, corrected. Units in previous Table 2 are in ppm.
5.Lines 199-200, page 6, "The increase in normalized mass was attributed to the oxidation of the fibers during heating stage. " So, what is oxidized during the heating stage?
Discussion on DSC and TG results was expanded. Following was added to the revised manuscript: "The increase in normalized mass was attributed to the oxidation of the fibers during heating stage. Oxygen is incorporated into fibers and forms metallic oxides, where oxidation of Fe2+ to Fe3+ occurs [28–30]. Moreover, this behavior is indicated by a broad peak up to the temperature of the onset of crystallization. After the first exothermic crystallization peak, the mass of the sample began to decrease."
6.Table 5, the change of modulus of elasticity and hardness with the variation of composition of those mixture, and why they changed like that should be stated in the text.
Following explanation was added to the revised manuscript; "
The hardness results obtained and the differences between individual fibers depend strongly on the amount of SiO2, Al2O3 and CaO. The chemical composition of the fibers given in Table 2 shows that the sum of all SiO2, Al2O3 and CaO is practically the same in all samples. However, their ratios are different and affect the fiber hardness. In sample S1, the proportion of SiO2 and Al2O3 is the highest, while sample S2 has the highest hardness, where a higher proportion of CaO was measured. Comparing the results with those of Deng et al. [17] , the sample with the lowest SiO2 content, which also has the highest CaO content, should have the highest hardness. The Al2O3 content is practically the same in samples S1, S2 and S4, while it is slightly lower in sample S3, which affects the measured hardness values."
Round 2
Reviewer 1 Report
The manuscript is greatly improved respect the previous version, but still contain some unclear aspects, typing errors and require English revision, as detailed below.
Line 50: This study consists…
Line 65-66: the fact that the starting composition in terms of diabase composition, slag composition and their reciprocal abundances are not known, is a major limitation of this paper, which aims at establishing a relationship between rock wool mechanical properties and composition of the starting mixture.
Line 73; extra bracket (or missing)
Table 1. Na is a major constituent of the starting mixing, and the fact that it could not be analyzed in the products is a great limitation of this study, which aims at establishing a relationship between rock wool mechanical properties and composition of the starting mixture.
Line 96: the melt…
Line 133: NIST-1d and NIST-88b (without space)
Line 145: as reference sample
Line 173-174: give the composition of the SiO2 alone also, for symmetry reasons!
Line: 173-180: the term “ceramic” is misleading. Ceramic is a general textural term which refers to synthetic materials with an equilibrium texture achieved by slow annealing at high temperature and made of pseudo-hexagonal grains (in two dimensions) with grain boundaries characterized by triple joints forming angles of about 120°. Allumina and silica are also major constituents of “ceramics”, as sanitary ware, tiles, pots, etc. None of these meanings applied to your materials.
Line 175-176: reword/revise English
Line 179: other components
Line 181: minute --> minor
Line 186: actually, Na2O can be detected by common XRF instruments, so why you could not detect it? That is a major limitation of your work because the starting mixtures contain major amount of Na2O, which is in addition a very powerful network modifier!
Table 2: The compositions of the products seem to differ mostly in terms of Fe2O3 and CaO. It would be worthy to focus the discussion more on this.
Line 204-205: which are the constituent phases that crystallize at these three temperatures? Diopside and anorthite? Whose the third peak is? In any case the temperatures do not correspond. Clarify and discuss any possible discrepancy.
Line 231-235: Above you report about the crystallization steps on heating, here on cooling. I suppose that the exothermic peaks you describe above are related to the crystallization diopside, anorthite and something else from the starting mixture components. Here you report again about crystallization, so I suppose there is a point on heating at which the previous phases melt, and then recrystallize on cooling. Is this correct? What are the phases that crystallize on cooling? Please clarify.
Line 236: Table 3 --> Table 4 and revise headers of the right columns
Line 281-283: revise text/English
Line 304: calculation. and
Lien 234: glass, basalt, minerals are they fibers? Please, revise
Author Response
We are grateful for the comments (italic), which have been addressed as follows:
Line 50: This study consists…
Corrected as suggested by the reviewer. Correction was applied in line 56.
Line 65-66: the fact that the starting composition in terms of diabase composition, slag composition and their reciprocal abundances are not known, is a major limitation of this paper, which aims at establishing a relationship between rock wool mechanical properties and composition of the starting mixture.
Although the exact chemical composition of the diabase and slag used is not known, the chemical composition of the mixtures obtained is given. The input mixtures are trade secrets and mineral wool manufacturers are reluctant to disclose the exact composition of the diabase and slags they use. However, based on the chemical composition given, the results could be repeated with a suitable mixture of basic oxides. Moreover, the mechanical properties are discussed with respect to SiO2-Al2O3-CaO variations and not in terms of diabase/slag variations.
Line 73; extra bracket (or missing)
Corrected as suggested by the reviewer.
Table 1. Na is a major constituent of the starting mixing, and the fact that it could not be analyzed in the products is a great limitation of this study, which aims at establishing a relationship between rock wool mechanical properties and composition of the starting mixture.
The reviewer is correct that it would be beneficial to obtain the Na content in the fibers produced. It is well known that Na2O plays an important role in breaking the SiO2 network by breaking silicate chains and forming Na+-O- bonds. However, comparing the obtained hardness results of samples S2 and S3, where the initial Na2O content is practically the same, the differences in hardness are quite significant. Therefore, the variation in hardness is due to the interplay of SiO2-Al2O3-CaO variations, as published in the cited references and discussed in the manuscript.
Line 96: the melt…
Corrected as suggested by the reviewer.
Line 133: NIST-1d and NIST-88b (without space)
Corrected as suggested by the reviewer.
Line 145: as reference sample
Corrected as suggested by the reviewer.
Line 173-174: give the composition of the SiO2 alone also, for symmetry reasons!
SiO2 reference measurement is provided in revised manuscript.
Line: 173-180: the term “ceramic” is misleading. Ceramic is a general textural term which refers to synthetic materials with an equilibrium texture achieved by slow annealing at high temperature and made of pseudo-hexagonal grains (in two dimensions) with grain boundaries characterized by triple joints forming angles of about 120°. Allumina and silica are also major constituents of “ceramics”, as sanitary ware, tiles, pots, etc. None of these meanings applied to your materials.
In the revised manuscript, the combined SiO2 + Al2O3 content is used instead of the previously used ceramic content.
Line 175-176: reword/revise English
Corrected as suggested by the reviewer.
Line 179: other components
Corrected as suggested by the reviewer.
Line 181: minute --> minor
Corrected as suggested by the reviewer.
Line 186: actually, Na2O can be detected by common XRF instruments, so why you could not detect it? That is a major limitation of your work because the starting mixtures contain major amount of Na2O, which is in addition a very powerful network modifier!
The reviewer is correct that XRF devices are capable of detecting Na these days. Unfortunately, we have the ThermoFisher Niton XL3t 900S + GOLDD, which is capable of detecting elements from Mg to U. We believe that the error of not considering Na2O when discussing hardness results is minimal, as explained in the response to the comment above (Table 1. Na is a major constituent...).
Table 2: The compositions of the products seem to differ mostly in terms of Fe2O3 and CaO. It would be worthy to focus the discussion more on this.
The reviewer is correct that the major differences are in CaO content and the minor differences are in Fe2O3 and SiO2 content. However, the hardness results have already been discussed in terms of the interaction of SiO2-Al2O3-CaO variations and compared with the literature.
Line 204-205: which are the constituent phases that crystallize at these three temperatures? Diopside and anorthite? Whose the third peak is? In any case the temperatures do not correspond. Clarify and discuss any possible discrepancy.
The endothermic peak is due to the glass transition temperature. Two exothermic peaks correspond to crystallization. From the literature, crystallization of diopside (CaO·MgO·2SiO2) was found to occur at about 865 °C and of anorthite (CaO·AI2O3·2SiO2) at temperatures around 1060 °C. Discrepancies in peak temperatures were attributed to different CaO/MgO ratios in the chemical compositions, citing the relevant literature.
Line 231-235: Above you report about the crystallization steps on heating, here on cooling. I suppose that the exothermic peaks you describe above are related to the crystallization diopside, anorthite and something else from the starting mixture components. Here you report again about crystallization, so I suppose there is a point on heating at which the previous phases melt, and then recrystallize on cooling. Is this correct? What are the phases that crystallize on cooling? Please clarify.
The term "crystallization" was not used correctly. When the materials in question are cooled, the correct term would be solidification. Solidification is used in the revised manuscript. The main interest in our study is only the solidification window and not the exact phase kinetics.
Line 236: Table 3 --> Table 4 and revise headers of the right columns
Corrected as suggested by the reviewer.
Line 281-283: revise text/English
Corrected as suggested by the reviewer.
Line 304: calculation. and
Corrected as suggested by the reviewer.
Lien 234: glass, basalt, minerals are they fibers? Please, revise
Corrected as suggested by the reviewer. Correction was applied in line 324 of previous revision.
Reviewer 2 Report
No more comments.
Author Response
The authors are grateful for the reviewers' comments. No comments were made by reviewers in this round of reviewing.